# Developing a Secure Low-Cost Radon Monitoring System

**DOI:** 10.3390/s20030752

**Published:** 2020-01-29

**Authors:** Alberto Alvarellos, Marcos Gestal, Julián Dorado, Juan Ramón Rabuñal

**Affiliations:** 1RNASA-IMEDIR, Computer Science Faculty, University of A Coruna, 15071 A Coruña, Spain; mgestal@udc.es (M.G.); julian@udc.es (J.D.); juanra@udc.es (J.R.R.); 2Centro de Investigación en Tecnologías de la Información y las Comunicaciones (CITIC), Campus de Elviña s/n, 15071 A Coruña, Spain; 3Centro de Innovación Tecnolóxica en Edificación e Enxeñería Civil (CITEEC), Campus de Elviña s/n, 15071 A Coruña, Spain

**Keywords:** radon monitoring, IoT security, alert system, open source

## Abstract

Radon gas has been declared a human carcinogen by the United States Environmental Protection Agency (USEPA) and the International Agency for Research on Cancer (IARC). Several studies carried out in Spain highlighted the high radon concentrations in several regions, with Galicia (northwestern Spain) being one of the regions with the highest radon concentrations. The objective of this work was to create a safe and low-cost radon monitoring and alert system, based on open source technologies. To achieve this objective, the system uses devices, a collection of sensors with a processing unit and a communication module, and a backend, responsible for managing all the information, predicting radon levels and issuing alerts using open source technologies. Security is one of the largest challenges for the internet of things, and it is utterly important in the current scenario, given that high radon concentrations pose a health risk. For this reason, this work focuses on securing the entire end-to-end communication path to avoid data forging. The results of this work indicate that the development of a low-cost, yet secured, radon monitoring system is feasible, allowing one to create a network of sensors that can help mitigate the health hazards that high radon concentrations pose.

## 1. Introduction

Over the past few years, society has begun to be more aware and more concerned about the impact that certain environmental elements may have on their health. This has led to the fact that certain control measures have begun to be taken due to the special awareness caused by every aspect related to health issues, directly or indirectly. In parallel, there has been a large proliferation of small sensors, wearables and control devices intended for measuring the most diverse variables and controlling the conditions that caused them.

This work aims to unite both aspects through a system for obtaining radioactivity environmental values based on the use of low-cost devices and a web application that will allow them to be easily consulted, as well as to store the obtained data for future analysis and facilitate the development of tools for decision making.

As a field of work, the monitoring of radon environmental levels was chosen due to its special incidence in Spain in general and in Galicia in particular, caused by its geology, which is strongly linked to granite, the main source of radon emissions [1,2].

Radon (Rn) is a colorless natural radioactive gas, with no smell or taste, which occurs in different forms, all of them with the same atomic number but different atomic mass, called isotopes [2]. As time passes by, the radon degrades by losing atomic mass¸ emitting radiation and transforming into another radioactive element. This is repeated cyclically until a stable state is reached, in this case until the radon is transformed into lead. The emitted radiation comprises alpha particles, beta particles and gamma rays, which can be a problem in terms of prolonged exposure to it [3].

Radon is measured in terms of its radioactivity (curies or becquerels). Both the curie (Ci) and the becquerel (Bq) indicate the amount of radioactive material that decays every second (1Ci = 37 billion Bq = 37 billion decays/second). The radiation dose of radon and its progeny (elements in which radon decays) is measured in terms of the energy they impart to the tissue (in units called gray or rem for general population exposure or levels for occupational exposure) [2].

The most common isotopes of radon are formed naturally from the degradation of uranium or thorium. Among those isotopes, the most common is radon-222 (222Rn), about 80% of the total, which emits alpha particles during its decay process, transforming into a polonium-218 atom (218Po), which also emits an alpha particle, transforming into a radioactive lead atom (214Pb). In the last stage of the radioactive decay of the radon progeny, a stable lead atom is formed, the latter being a non-radioactive element. The half-life of 222Rn is 3.82 days; that is, after 3.82 days, the initial amount is reduced by half. However, both uranium and thorium have very high half-lives (4470 and 14,000 million years, respectively) so that their presence, and consequently that of radon, will continue to exist indefinitely at virtually the same levels as today. Therefore, the greatest emphasis should not be placed on its complete elimination, but on its adequate detection and adequate response to minimize prolonged exposure to it.

Because the radon progeny often adheres to dust, the greatest probability of exposure is due to the simple act of breathing the suspended dust, since the radon progeny is found in almost all of the air [1]. Depending on the size, the radioactive particles can be deposited in the lungs and deliver a dose of radiation to the lung tissue. The radon concentration in the atmosphere is usually measured in Becquerels per cubic meter (Bq/m^3^) or picocuries per liter (pCi/L), with a pCi/L being equivalent to 37 Bq/m^3^ [1,2].

The actual levels of radon in the open air are generally very low (0.4 picocuries/L or 14.8 Bq/m^3^ is the average radon activity in the air in the United States), but may vary according to the location and soil geology. The actual levels also vary depending on weather conditions such as precipitation and temperature reversals.

In indoor air, for example at home, schools or in office buildings, radon levels and their progeny are generally higher than outdoors. The fluctuations are important but can range from 10 Bq/m^3^ to 10,000 Bq/m^3^, according to reports from the World Health Organization [4]. The type of construction can affect radon levels; however, there may be high levels of radon in homes of all types—old or new homes, homes with air currents, homes with insulation, and homes with or without a basement. With radon having a higher density than air, its concentration is higher the lower the height of the house compared to the upper floors. Atmospheric conditions, local geology, building materials, ventilation habits, and the way housing was built are some of the factors that affect radon levels in homes [1,2].

Radon typically moves through the earth into the air and passes into homes through cracks and other holes in the foundations, due in part to convection flows. It can also be found in the water coming from underground flows and/or wells due to its contact with areas that are rich in non-decayed elements. There are really no reliable techniques valid to know the radon level in a location, except for measurements, if possible, which are prolonged in time and not punctual.

Radon has been declared a human carcinogen by the United States Environmental Protection Agency (USEPA) and the International Agency for Research on Cancer (IARC) [3].

In Spain, the pioneering studies regarding the taking of environmental measurements date from the end of the 1980s [5]. These studies highlighted the high concentrations of radon in several regions, including, among others, Galicia.

The European Union (EU) has indicated in its guidelines a reference level of 300 Bq/m^3^ of annual average exposure that should not be exceeded in homes and centers or work sites (E2013/59/EURATOM [6]). This same guideline makes explicit the need for member states to include in their technical building codes the problem related to radon detection and mitigation. On a comparative level, the threshold value established by the US Environmental Protection Agency is 148 Bq/m^3^ (4pCi/L) [7] and the threshold set by the World Health Organization is only 100 Bq/m^3^ [4].

Recently, the Nuclear Safety Council made a map of exposure to residential radon, shown in Figure 1, which shows that the areas that are most at risk are Galicia, the south and west of Castilla y León, the North of Extremadura and the Community of Madrid. One of the most probable causes derives from the mainly granitic soils of these areas, therefore with significant amounts of uranium present in the granite.

In Figure 2, extracted from the Cartography of the Radon Potential of Spain, the possible routes of entry of radon in a house are schematized.

As we can see in Figure 1, most of Spain’s radon levels are above the 100 Bq/m^3^ base level of the WHO handbook [4], and approximately 50% of Spain’s radon levels are above 200 Bq/m^3^. This work was carried out in Galicia, in the northwest of Spain, where the indoor radon level could be above 400 Bq/m^3^. The WHO handbook states “...*establishing a national annual average concentration reference level of 100 Bq/m^3^, but if this level cannot be reached under the prevailing country-specific conditions, the reference level should not exceed 300 Bq/m^3^*”. Due to the high radon levels of Spain, our system uses the 300 Bq/m^3^ limit of the WHO handbook as the radon gas level threshold.

## 2. Research Problem

The main objective of this work was to prove the feasibility of a low-cost radon gas monitoring system that would help to reduce the radon gas concentration levels. We attempt to prove it by creating a low-cost system, based on open source technologies, to monitor radon levels and implement both a basic radon level prediction and an alert system to inform an interested party of the presence of radon gas by predicting its concentration before reaching a dangerous level, allowing one to take measures to avoid the raise of the radon gas level. 

As a secondary objective, and given that high radon concentrations cause health problems [3], we need to secure our system to minimize the impact that either attacks or their effects could have on the health of the people where the radon levels are measured due to data forging, which introduces false information into the system.

### 2.1. Radon Detection

When making measurements to determine radon concentration, there are different options, both short and long term. All of them usually provide reliable measures, but it must be taken into account that radon concentration will be conditioned by factors such as temperature, pressure and/or humidity. For this reason, long-term measurements are recommended, to mitigate their punctual influence as far as possible.

Depending on the sampling time, three methodologies or approaches can be discussed:**Instantaneous methods:** the measurement is carried out based on samples of a short period of time (between approximately 1 and 20 min). Its main advantage is the speed concerning the obtainment of results, and that they are possible through the use of low-cost measurement systems. These methods are commonly used in initial approaches.**Continuous reading methods:** these are based on the passage of a continuous and constant flow of air through a detector, which performs the measurement over long periods of time. They are very reliable methods that provide a great deal of information, essential for the analysis of fluctuations in concentration. They are more expensive than the previous ones.**Integrated methods:** they measure concentration over days, or at most several months. They are the most used since they are considered suitable for carrying out inspection studies prior to the decision to take corrective measures.

Regarding the type of equipment used to carry out the measurements, we can talk about scintillation cells/pulse ionization chambers, solid trace detectors or activated carbon detectors:**Scintillation cells/pulse ionization chambers:** these consist of metallic cylinders, with a transparent end, and coated inside by a uniform layer of zinc sulfide that is activated against alpha-type emissions. They allow for online measurements through a photomultiplier tube to determine the concentration of radon in the air. The pulse ionization chamber technology is similar to that of scintillation cells in the sense that it also uses a metallic cylinder, probes and an amplifier to detect the secondary charges which were generated from collisions between the air in its interior and the α-particles caused by radon or radon’s progeny.**Solid trace detectors:** these are passive systems that are based on the use of materials in which the traces of radon and its descendants are impressed after a period of exposure, generally prolonged. These are subsequently analyzed in the laboratory to determine the number of traces under the microscope and, therefore, the radon concentration.**Active carbon detectors:** like the previous exampless, these are also passive detectors. These are based on the ability of activated carbon to retain radon. Unlike the previously mentioned systems, they work by measuring the amount of gamma radiation emitted in periods of generally less than one week.

### 2.2. Analysis of Previous Work

After reviewing the state of the art on radon gas monitoring devices, whole systems, and the related academic work, we arrived at the conclusions that follow.

Several commercial devices measure radon gas concentration, but either they are too expensive for the average householder (mid to high-end devices are in the thousands of euros) [8], or they are cheap but lack characteristics such as internet connectivity and need a specific desktop software for collecting the data [9], do not allow the user to send the data to a specific backend and the data gathering is not easy [10,11,12] or are slow because they need to be placed for a long time (several days to months) in the same location and then analyzed in a laboratory to measure the radon exposure of the device [13].

Regarding the type of connectivity of the devices, some devices allow the user to collect the data via USB [9] or Bluetooth [12] or send them to the manufacturer backend via Wifi [12]. There is one academic work that created a system (but not a production-ready product) that used a commercial device and added LoRaWAN connectivity to send data to their system [14]. 

These technologies do not properly cover the characteristics of the northwest of Spain. Not all households and public spaces have WiFi connectivity and Galicia has an old population that is not technology savvy, so we need a device that requires minimal configuration. Although LoRaWAN could be the solution for such scenarios, it requires one to have (or to install) a LoRa gateway in the region where the device is going to be used and Spain lacks a proper LoRa network deployment. We consider that Sigfox technology is more appropriate for such cases. As LoRaWAN, Sigfox is one of the so-called low power wide area networks (LPWANs), which also uses an ultra-narrow frequency band to achieve high levels of coverage while consuming low levels of energy, but the main difference is that Sigfox, contrary to LoraWAN, does not require a gateway in the region where the device is going to be used.

There has also been some research done on mapping radon for a given area [15,16] but this work used a commercial device [12] to manually gather the data, instead of creating a new, and modifiable system that could be used in our work.

There was also some research done in the same region as this work, for instance [17] where the researchers bypassed a commercial device (Safety Siren Pro Series 3 Radon Gas Detector), used for instantaneous or continuous measurements of radon gas concentration, to relay the information to their system via a WiFi SoC. However, it lacked the security aspect we think such a system should have and we think just WiFi connectivity is not the appropriate approach, as previously explained.

As we said previously, we must take into account that radon concentration will be conditioned by factors such as temperature, pressure and/or humidity. The commercial radon measuring devices presented just measure radon, and the previous work previously explained did not gather these variables.

High radon concentrations pose a health risk and, since security is one of the largest challenges for the internet of things, we think that security has to be taken into account when developing a radon monitoring system. The works previously mentioned lacked in such aspects.

The work we present in this paper resolve these issues by creating a fully-fledged device that measures radon concentration, temperature, humidity, barometric pressure, and air quality, both with WiFi and Sigfox connectivity and a secure backend that is responsible for managing all the information the devices collect, predicting radon concentrations, issuing alerts if a high radon concentration is predicted, and showing the data and alert status on a web.

## 3. Materials and Methods

If we want to control the radon levels in a habitable space, we need to renew the air the space contains. In regular size spaces, this can be done by opening a window or door, allowing the clean air to fill the space. In bigger or more isolated spaces, we would need to use an air flow control system that can process the volume of air of the space in a reasonable time (1 h), replacing it with clean air.

The system we propose has as its objective to inform the user, ahead of time, that a regular-sized space is going to have a high radon concentration. This will allow the people that occupy said spaces to avoid the rise in radon concentrations.

To achieve these objectives, the radon control system uses radon sensors to monitor radon levels and uses the MQTT, email and visual alerts to inform interested parties.

### 3.1. System Architecture

The system was designed to continuously sample radon levels in close to real-time; a real-time system would just detect a particle collision, while this measure needs to be integrated over time (usually 10 min to 1 h), extrapolated for a longer time frame and compared to the limit used in the system (regulatory limits are usually established for annual continuous levels).

To create a close to real time system, we needed to use an instant method that reads the radon level without the need of human interaction for analyzing the samples, so we chose an instant system based on an ionization chamber.

The radon control system architecture is shown in Figure 3. The system is comprised of two parts: the devices (sensors) and the server (backend).

The devices send data periodically to the server. The server stores the data and makes a prediction of the radon level for each device. If a high radon concentration is predicted for a given device, the server sends an MQTT message to an MQTT topic associated with the device. The MQTT message also triggers the sending of an email to the addresses associated with the device. Once the radon concentration level is below the threshold, the server sends a message indicating that levels are back to normal (via a MQTT message to the device MQTT topic and the corresponding email message).

The implementation of each component of the system and the security implementation are detailed in the ensuing sections.

### 3.2. Devices (Sensors)

The main objective of this work was to create a secure system to monitor radon levels and implement both a basic radon level prediction and an alert system to inform the interested parties of future high radon levels. The devices developed for this project were designed around a radon sensor. As we previously said, we chose an instant sensor based on a pulsed ionization chamber: the RD200M sensor [18]. Its main characteristics are:Sensitivity: 0.81 cph at 1 Bq/m^3^.Measurement range: from 7.4 to 3.700 Bq/m^3^.Precision: ±10% at 370 Bq/m^3^.Default 10 min collision integration period.Each sensor is individually calibrated.It has a built-in vibration sensor for preventing errors in radon detection.

As we previously explained, our system uses a 300 Bq/m^3^ threshold for issuing the alerts. The RD200M sensor can read radon concentration from 7.4 to 3.700 Bq/m^3^, so its characteristics make it suitable to use in our system.

A secondary objective was to use the system to collect a high volume of data to be able to create better predictors in the future. We think that there are other environmental variables that could influence radon levels. In order to collect data and, in future works, verify this assessment, we incorporated several environmental sensors, along with the radon sensor, in the devices developed for this project:The BME280 sensor [19]: a digital sensor for measuring relative humidity, barometric pressure and ambient temperature.The CCS811 gas sensor solution [20]: a digital gas sensor for monitoring indoor air quality.

In summary, we have sensors to collect the following data: radon concentration, temperature, humidity, barometric pressure, and air quality. The data from all these sensors are collected by a processing unit and sent to the server developed in this work. The schema of the devices is shown in Figure 4. The processing units were developed using the Arduino mkr microprocessor family. These microprocessors are based on the ARM Cortex-M0+ CPU (running at up to 48 MHz). We chose this platform for the prototypes because the ARM Cortex-M0+ processor is a high-performance processor and it is energy-efficient at the same time. This high-performance processor is needed to process all the data from the different sensors and to secure all the communications with the server (explained in a later section).

The processing unit samples each sensor but the radon one every 5 s, and the radon sensor every 10 min. After receiving the data from the radon sensor it sends the data to the server (this sampling period is recommended by the radon sensor manufacturer in order to integrate the measured collisions for a reasonable amount of time).

As we said before, the system’s devices use two different communication technologies: WiFi and Sigfox. The WiFi device is used in locations where a WiFi network is available and we have access to it. WiFi technology is widely known and we will not expand on its inner workings, but we will explain how we secured the WiFi communications with the server in the Security section.

The other device uses the Sigfox Communication Network. Sigfox is one of the so-called low power wide area networks (LPWANs), which uses an ultra-narrow frequency band to achieve high coverage while consuming low energy. Sigfox is based on a star topology and, for the region where the devices were tested (northwest of Spain), it uses the configuration shown in Table 1.

The transmission speed and maximum data exchange are lower than other LPWANs such as LORAWan and NBIOT [21], but the Sigfox network provides broader coverage. This is especially relevant for environmental monitoring because it allows using the devices in regions where the access to other communication networks is not possible, for instance, there are some rural areas in the northwest of Spain without internet access, hence using WiFi is not possible.

Sigfox communications use a public band (868MHz). The European Telecommunications Standards Institute (ETSI) regulations stipulate that devices operating on a public band can only transmit 1% of the time over the course of 1 h. In Sigfox, this translates into 6 messages, of 12 bytes each, per hour, or 144 messages per day. This restriction is not imposed as a hardware limitation but it is a gentleman’s agreement. Despite the low data exchange rates, it is sufficient for sending the data that the devices developed in this work collect: our devices send data every 10 min, thereby we need to send 144 messages per day.

In both cases, the devices send the data to a server that stores the data, makes a prediction and sends an alert if necessary. The structure and inner workings of the server are explained in the following section.

### 3.3. The Server (Backend)

The backend functionality was implemented using a collection of virtual servers (see Figure 3). The virtual machines were created and managed using the Proxmox Open-Source Virtualization Platform (Proxmox VE [22]). Proxmox VE is an open-source platform for enterprise virtualization that integrates a KVM hypervisor and LXC containers, software-defined storage and networking functionality on a single platform. It allows the managing of high availability clusters and disaster recovery tools through a web management interface.

All the communications that reach the Proxmox server are tunneled through a virtual machine where pfSense manages all the connections. PfSense is an open-source firewall, VPN, and router. It can be used to address firewall, routing and VPN server needs. It is also widely deployed to address secure networking needs and includes load balancing, traffic shaping, web content filtering, a transparent caching proxy, and reverse proxying among its features [23]. In our case, we used the pfSense firewalling capabilities to filter the connections that reach the server and its reverse proxy capabilities to route the data to the server that processes it (implemented using the HAProxy module, an open-source software that provides a high availability load balancer and proxy/reverse proxy server for TCP and HTTP-based applications).

The pfSense node filters the data and redirects the filtered data to the data processing node. The data processing node is implemented using Node-RED, a programming tool for wiring together hardware devices and APIs [24]. Node-RED provides a browser-based editor that makes it easy to wire together processing flows using predefined nodes that can be easily deployed to its runtime.

In summary, all the data that reach the backend are processed as explained in the pseudocode of Text 1.

Text 1backend data processing pseudocode.*In* ProxMox:   tunnel all data to pfSense*In pfSense*: **if** protocol in [https, MQTT, email] **then**  **if** data destination domain correct **then**   send data to processing node  **else**   block data **else**  block data*In the * processing node: data ← listen in http endpoint send http response    store data in database    publish data on web    prediction ← predict_radon_level    **if** prediction >= threshold **then**  **if** not already in alert **then**    send alert    store alert in database **else**  **if** not already normal **then**    send back to normal message    store message in database

The database schema is shown in Figure 5. We used several sensors (devices) whose information is stored in the sensor table. Each device’s measures are stored in the measure table. We used several actuators (means of sending alerts/back to normal messages), such as email and MQTT messages, that are stored in the actuator table. Each time a message is sent, it is stored in the action table.

Step 4a merits a more detailed explanation—after storing the data for a given device, the server predicts the radon level for the next hour. If a high radon concentration is predicted for the given device, the server sends a message to the actuators associated with the sensor to inform the interested parties. When the radon level is lower than the threshold, the server sends a message via the actuators to indicate that the radon levels are back to normal. The estimation the server makes is done using linear regression based on an exponential moving average of the radon level slope (see Equation (1)): each time a radon value is received (every 10 min), the moving average of the slope of the radon level is calculated (using α = 0.85). Using the current radon measure and the averaged slope, we project the radon level 1 h into the future (60 min). If the value we obtain is above the threshold (set to 300 Bq/m^3^) we log an alert into the database and send it (via MQTT and email).
(1)$$\begin{array}{c} slope_{t}=\left(radon_{t}\text{-}radon_{t-1}\right)/10\\ slopeAvg_{t}=\alpha *slope_{t}+\left(\alpha -1\right)*slopeAvg_{t-1}\\ radon_{t+60}=radon_{t}+60*slope_{t} \end{array}$$

## 4. Security

Given that high radon concentrations cause health problems [3], we needed to secure our system to minimize either attacks or their effects. Figure 6 represents the main parts of our system, the data flow between them and how they are located in the different areas of the communication network. The more critical devices are included in more restricted areas. Two elements centralize and secure the access to this private zone: the pfSense node and the processing node (based on Node-RED as previously explained). In the following paragraphs, we explain the possible problems the system could face and the security approaches we used in the different layers.

The public devices of the system, included in the DMZ, are the most likely to be attacked. Programming-related attacks, such as SQL-injection and cross-site scripting, were taken into account using secure programming practices and robust frameworks (like Node-RED) when we developed the system.

Due to the type of information managed by the system, special importance must be given to securing measures to avoid data forging attacks (see Table 2)—if the system is not secure, a hacker could intercept the sensors data or the sending of a message and block them or replace them with false information or place the sensor in a new location with a different radon concentration level than the original placement. From all possible attacks, the most dangerous scenarios depicted in Table 2 would be Scenarios 1, 4, 5 and 6.

In Scenario 1, a hacker changes the device data and sets (always or most of the time) the radon level the device sends to the server to a value under the threshold that we use to send the alerts. This attack would only be detected if someone is visualizing the radon levels the physical device is showing and realizes that the radon concentration levels the server stores are different than the measured ones.

Scenarios 4 and 5 affect the alert messages (MQTT and email messages), i.e., the information the server sends to the interested parties advising them to take measures in order to avoid the raising of radon levels. In Scenario 4, a hacker impersonates the server and sends a message indicating that the radon levels are back to normal (this attack would be more effective if sent some time after the server send a message indicating future high radon levels). In Scenario 5, the hacker intercepts all messages, so the interested parties are unaware if a message was sent. This attack would only be detected if someone is actively visualizing the radon levels and realizes that the levels are high and an alert is not issued.

The danger of all of these scenarios is that there could be a high radon concentration and we would not send an alert because we are unaware of it. If the alert is not sent, the interested parties would not take any action towards mitigating the radon levels. If this situation persists in time, it could have negative health consequences.

In order to minimize the probability of each one of these scenarios occurring, we need to secure several paths of the system’s communication infrastructure. In Figure 1, we can see the communication paths: the green ones are the already secured paths, the red ones are not secured by default and need to be secured ad-hoc, and the orange ones can be easily secured with the tools that implement the corresponding functionality (MQTT or email server). We will explain how we secured the paths that needed to be secured ad-hoc. They can be classified into two categories: communication paths and configuration paths.

Currently, the system’s sensor devices use two different communication technologies—WiFi and Sigfox—but we plan to incorporate and test more communication technologies in the future. When securing the communication paths, each technology has different requisites, but we tried to create a system that could be secured as independently as possible from the communication technology used.

We want our system to use HTTPS for communication, so both endpoints have to be able to use SSL to encrypt the data using certificates (devices and backend). There are two nodes in the backend where we could encrypt the communication: in the pfSense node or the Node-RED node. We chose to use the former via the load balancer (HAProxy), using the technique known as SSL offloading. This relieves the Node-RED server of the processing burden of encrypting and decrypting traffic sent via SSL, freeing its resources to process data. The processing is offloaded to the pfSense server that encrypts the communication with the devices while using non-encrypted communication with Node-RED (the communication between HAProxy and Node-RED takes place in the Proxmox virtual network, so it does not need to be additionally secured).

To use SSL certificates in pfSense, we used the ACME package. This software interfaces with Let’s Encrypt [25] (using the ACME protocol [26]) to handle the certificate generation, validation, and renewal processes. Certificates from Let’s Encrypt are domain validated. This validation ensures that the system requesting the certificate has authority over the domain in question. This validation can be performed in several ways, such as by proving ownership of the domain’s DNS records or hosting a file on a web server for the domain (we use the latter). By using a certificate from Let’s Encrypt for the data communication between the devices and the backend (more precisely between the devices and the virtual node running pfSense), the devices will trust the certificate and the connection will be encrypted without the need for manually trusting an invalid certificate in each device.

Up to this point, we have secured one endpoint: a backend with HTTPS (with SSL certificates management) and firewalling capabilities that reverse proxies the data to the processing node (see Figure 7). We need to secure the other endpoint—the devices. The WiFi devices send the data directly to the backend, so we only need to secure one hop in the communication path. The Arduino model used for the WiFi devices is the Arduino Mkr WiFi 1010. This model uses an ATECC508A CryptoAuthentication device—a cryptographic co-processor with secure hardware-based key storage that allows us to perform high-speed public key (PKI) algorithms [27]. Using this capability (and the appropriate library), we were able to send data securely via HTTPS (using SSL) directly to the backend.

The Sigfox devices–backend communication path comprises two hops: the one between the emitting device and the Sigfox Core Network and the one between the Sigfox Core Network and our backend.

The first hop (the Sigfox network between the devices and the Sigfox Core Network) is already secured by design, shielding the Sigfox devices from the internet by a very strict firewall. The Sigfox devices do not connect directly to the Internet, do not communicate using internet protocols (TCP/IP) and are not permanently connected to any network or base station. The devices operate predominantly offline. When a device requires data to be transmitted to the Internet, it broadcasts a radio message that is picked up by several base stations and is then transferred to the Sigfox Core Network and stored in the Sigfox Cloud service. This network architecture effectively provides an air gap, making it impossible to maliciously access an endpoint via the internet.

Regarding the second hop, the Sigfox Cloud service can be configured to send each incoming message to a given callback URL, our backend in this case. Since we already gave the backend the ability to process HTTPS messages, we only have to configure the HTTPS endpoint of our backend in the Sigfox Cloud service as the callback URL to secure this hop.

The last path we need to secure is the path used to send the configuration data to the devices (the configuration path). Sigfox devices connect directly to the Sigfox network (they broadcast a radio message that is picked up by several base stations) without requiring any configuration. The WiFi devices need the SSID and password of the WiFi network we want to connect them to. If this procedure is not secure, it could be used in Scenario 6 (Table 2) to change the location of the device to a new one with low radon levels, i.e., we could be having high radon levels without knowing it (and consequently not acting on it). To store the SSID and WiFi password, we use the NTAG I^2^C plus [28], an NFC Forum Type 2 Tag (passive) with I^2^C interface. When an NFC-enabled device (such as a smartphone) is close to the NTAG antenna, it connects with the tag and creates an electric field that can power the NFC chip allowing two-way data transfer between the NTAG and the NFC enabled device. Once the NTAG is powered, a microcontroller (MCU) can communicate with the NTAG using the I^2^C bus. The data transfer between the NTAG and the MCU can be either via the NTAG SRAM (pass-through mode, used for fast data exchange) or it could be via the NTAG EEPROM (used for non-volatile data storage). This schema is illustrated in Figure 8.

This device allows us to use an NFC-enabled device to send the WiFi configuration data to the MCU (the Arduino Mkr used in the devices). The configuration data consist of a user, password, SSID, and WiFi password. When the device is booting, it checks for a few seconds if new data can be read from the NTAG via I^2^C. We used the pass-through option of the NTAG; the data flow from the NFC interface through an SRAM buffer to the I^2^C serial bus interface (or vice versa).

The device’s MCU has a hardcoded user and password for administration purposes. When a new configuration is received from the NTAG, the MCU checks if the user and password are correct. In such a case, it stores the new SSID and WiFi password in the MCU EEPROM and uses it to connect to the WiFi in the current and in future reboots.

## 5. Results and Discussion

The main objective of this work was to create a secure and low-cost radon monitoring and alert system, based on open source technologies to prove its feasibility. We achieved this objective by creating devices, a collection of radon level, temperature, pressure, humidity, and air quality sensors with a processing unit and a communication module, integrated using a 3D printed case (see Figure 9 left, where the assembled device is shown), and a backend, a collection of servers responsible for managing all the information the devices collect, predicting radon concentrations, issuing alerts if a high radon concentration is predicted, and showing the data and alert status on a web (see Figure 9 right).

High radon concentrations pose a health risk and, since security is one of the largest challenges for the internet of things, we secured the entire end-to-end communication path using several tools and techniques to avoid data forging that could create a health hazard.

As a result of this work, we have created a low-cost radon level monitoring, prediction and alert system that gives radon level measures (and the other environmental variables explained before) in almost real-time, issuing an alert indicating that we need to take measures to reduce the radon levels whenever the system predicts a high radon concentration.

The system was created to inform, using alerts, that a high radon level is predicted for a given location. This enables taking action to avoid future high radon levels. Currently, the action has to be taken by a human operator. This action is usually opening a window or turning on an airflow control system. Figure 10 shows the radon levels in two scenarios for the same location. This location was used as one of our testing sited due to its particularly high radon concentration. In the first scenario (Figure 10 up), the alert system was not activated—the system just stored the radon measures, showing the natural radon level of the testing site. In the other scenario (Figure 10 down), the alert system was activated and a human operator was in charge of turning an airflow control system on or off (with each alert/”back to normal” message). Comparing these two scenarios, we can see that the system can satisfactorily be used to significantly lower the radon levels. These results also indicate that the selected radon sensor and the approach used when developing the system were appropriate for the intended usage.

The system created in this work and the data collected by it are set to be the bases for several future developments:Create a full radon mitigation system: we plan to create a device that can control an airflow control system. Pairing the radon control system with these devices, we can automatically activate the airflow when a high radon concentration is predicted. This would be more efficient than relying on users to take measures to mitigate the radon concentration when an alert is issued.Use an AI-based predictor: the current radon level prediction is based on a linear regression using an exponential moving average of the radon level slopes. Using an AI model that uses all the variables the devices are collecting (radon level, temperature, pressure, humidity, and air quality) could create a better predictor that could help to minimize the radon levels.

## Figures and Tables

**Figure 1 sensors-20-00752-f001:**
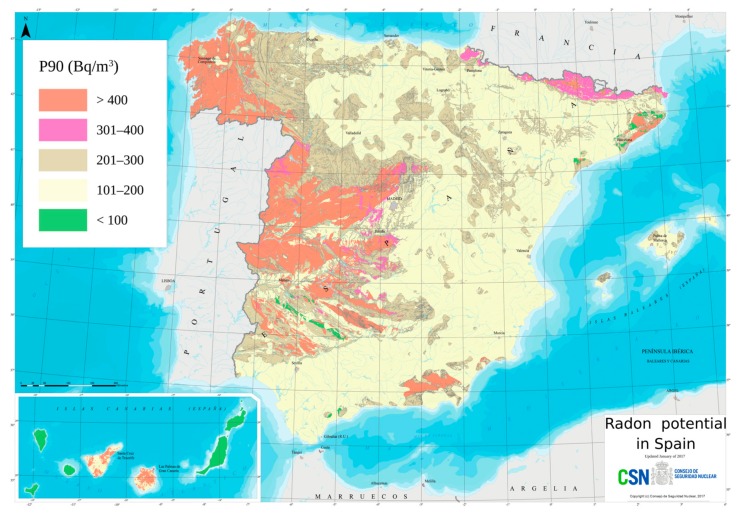
Map of radon potential in Spain (Nuclear Safety Council). Available at https://www.csn.es/mapa-del-potencial-de-radon-en-espana. The radon potential of an area is the 90th percentile (P90) of the radon distribution of buildings in an area. For example, P90 for a concentration of 300 Bq/m^3^ indicates that 90% of buildings have concentrations below that threshold, while the remaining 10% exceeds them.

**Figure 2 sensors-20-00752-f002:**
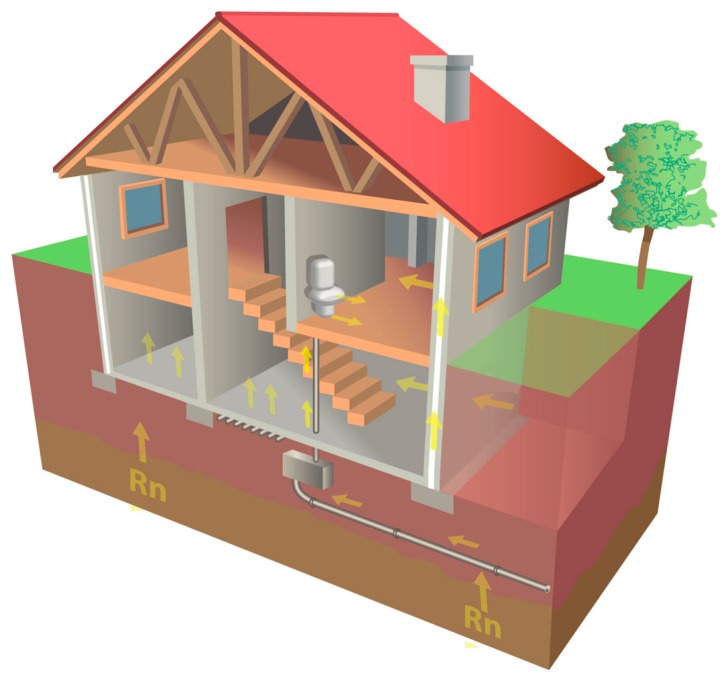
Radon accesses to a home: access through cracks in the subsoil, accumulation in lower floors and/or basements due to poor ventilation, through water pipes that are contaminated in the subsoil and transfer radon to higher levels, where it is filtered through taps, toilets and/or showers.

**Figure 3 sensors-20-00752-f003:**
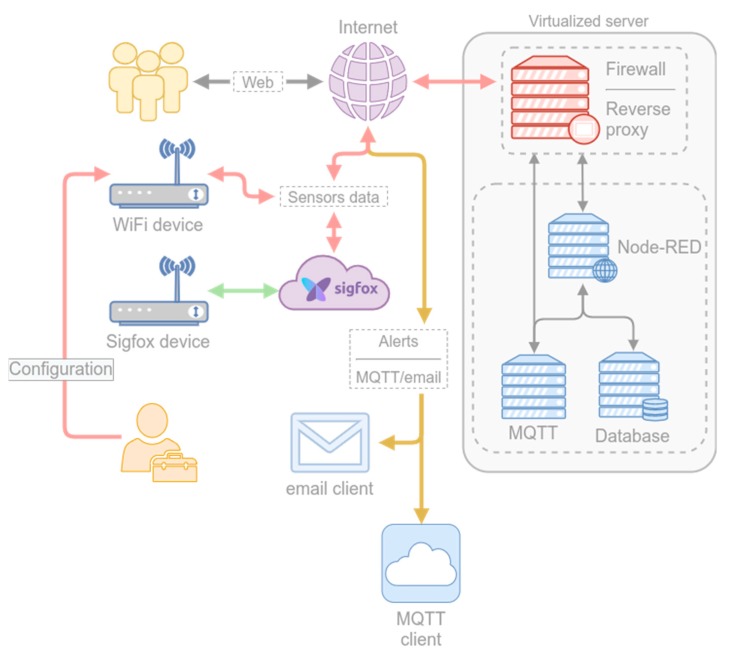
Architecture of the radon detection and alert system.

**Figure 4 sensors-20-00752-f004:**
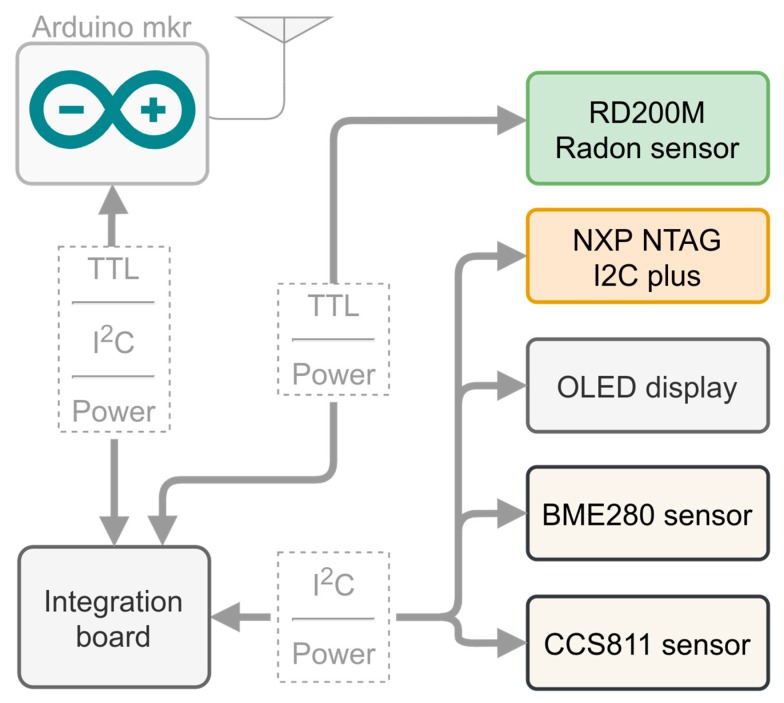
Sketch of the devices developed: processing unit and antenna, integration board, RD200M radon sensor, OLED screen, NXP NTAG I^2^C plus, CCS811 sensor and BME280 sensor.

**Figure 5 sensors-20-00752-f005:**
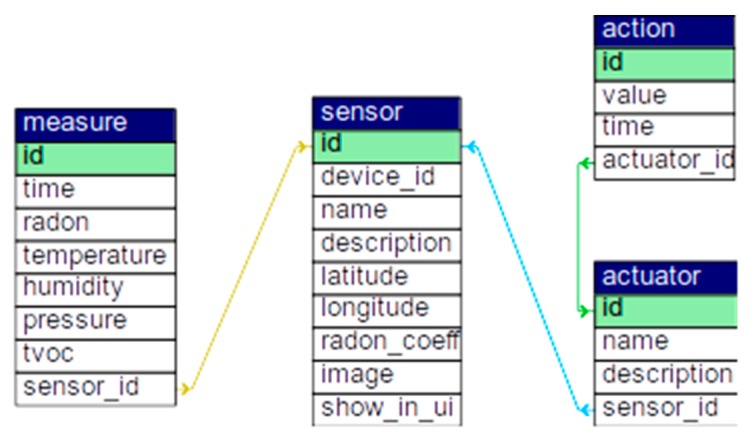
Database schema.

**Figure 6 sensors-20-00752-f006:**
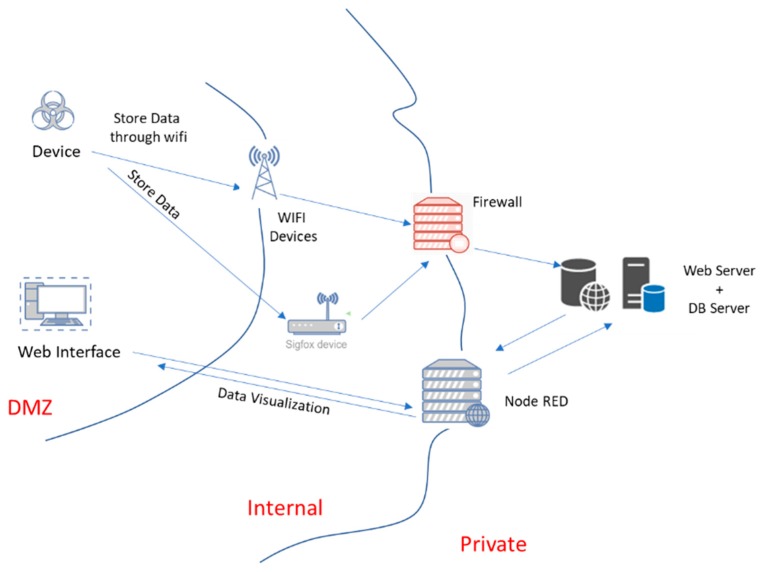
Thread model of the radon monitoring system.

**Figure 7 sensors-20-00752-f007:**
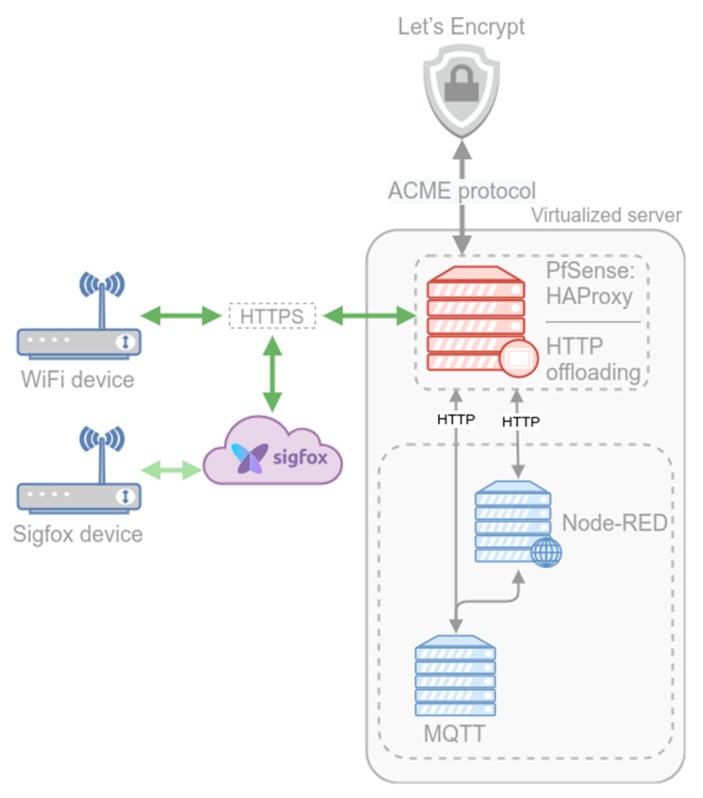
Secured communication schema.

**Figure 8 sensors-20-00752-f008:**
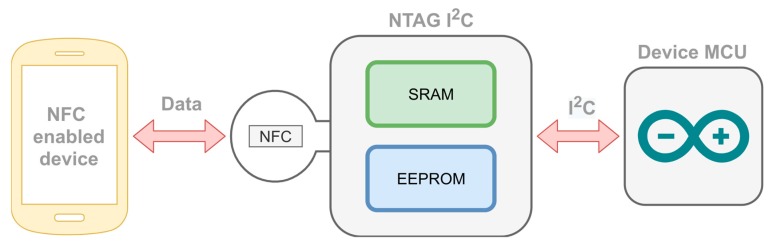
WiFi configuration of the device via NTAG I^2^C.

**Figure 9 sensors-20-00752-f009:**
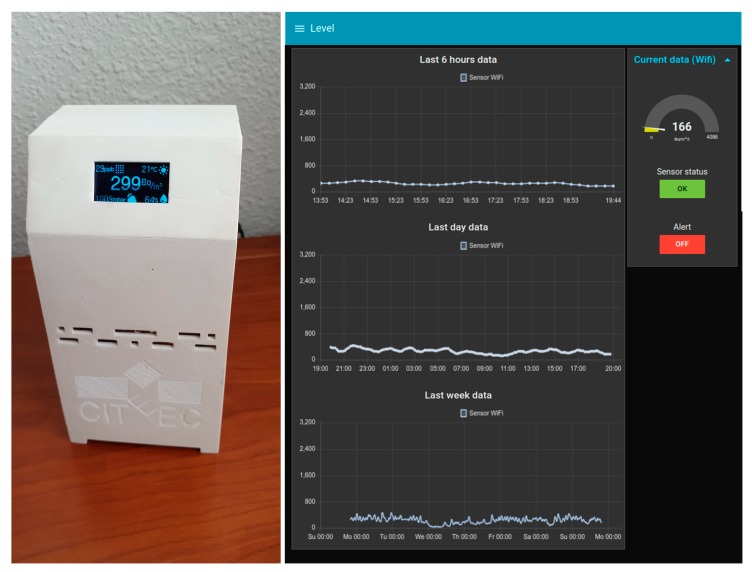
Device photo (**left**) and system web interface (**right**).

**Figure 10 sensors-20-00752-f010:**
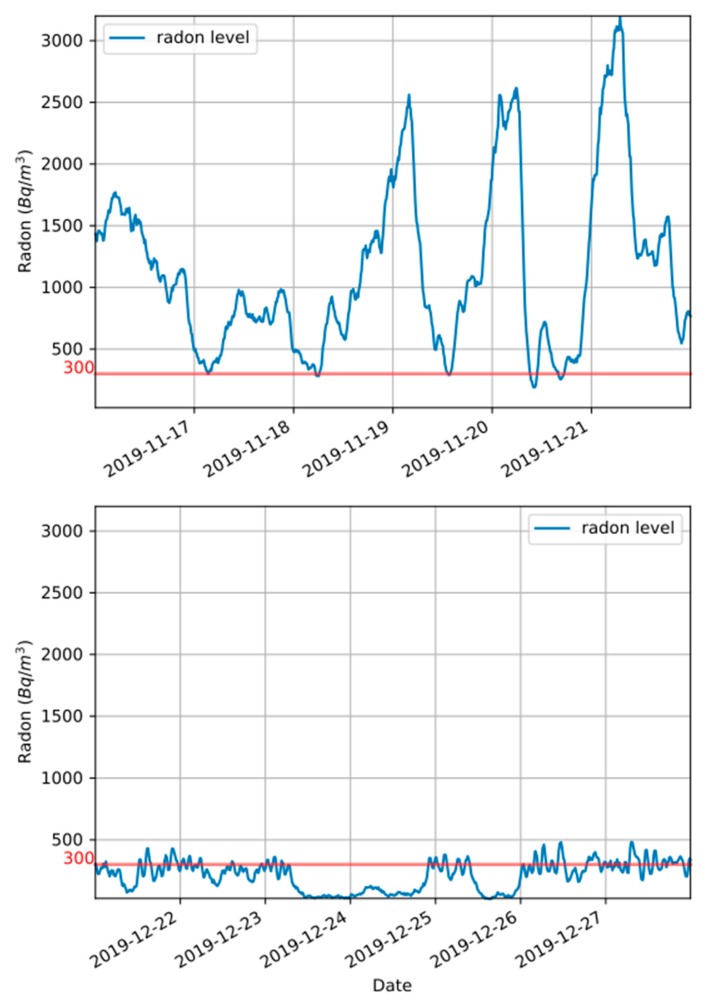
Radon levels for the same location. **Up**: alert system activated. **Down**: alert system activated and air flow control system operated by human.

**Table 1 sensors-20-00752-t001:** Sigfox characteristics for Radio Configuration 1 (the one used in Spain).

Center Frequency	Bandwidth	Data Rate (bit/s)	Recommended Effective Isotropic Radiated Power	Constraints
868.130 MHz	200 kHz	100 bit/s	16 dBm	140 messages/day

**Table 2 sensors-20-00752-t002:** Data tampering possibilities.

Data	Action 1	Action 2
Sensor data	Set the value below the threshold ^1^	Set the value above the threshold ^2^
Messages	Fake server messages: High levels detected ^3^ Levels back to normal ^4^	Intercept MQTT/email ^5^
Sensor location	Change the device to a new location with a low radon concentration ^6^	Change the device to a new location with a high radon concentration ^7^

^i^ ≡ scenario i.

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
