# Peer review of "Developing a Secure Low-Cost Radon Monitoring System"

_sensors, 2020, doi:10.3390/s20030752_

Round 1

Reviewer 2 Report

The manuscript presents the design of a low cost system that is able to monitor radon concentration levels as well as other variables of interest using several soft and hard technologies. Even though the approach is interesting and relevant from a technical point of view there is a main drawback with the manuscript: it lacks of a validation method. I strongly suggest to include a methodology to validate the performance in terms of standard metrics in order to evaluate both the security and the monitoring capabilities of the system. Without a validation method, the study is just a integration of several technologies that authors claim that works properly with no scientific evidence.

By the other hand, the title of the study does not reflect the fact that the system designed is a monitoring system not a control one. Authors do not present any control system issue developed in the study at all (open loop, closed loop).

In order to enhance the content of the manuscript there are several considerations that I suggest to review:

Pp 1, line 32 use "etc" or "so on" or "among others", instead of ellipsis (dots).

Pp 3, line 50, “transforming the process” is confuse, due to the statement is related to radon transformation not other process transformation.

Pp 7, line 194, why approximately, an explanation is required.

Pp 7, line 202, "said device" or "such device"?.

Pp 9, Fig 4 is messy, please re edite using a CAD software such as Fritzing (looks like it was used).

Pp 11, line 324, Fig 5.

Pp 12, line 333, it is more apropiate "estimation" instead of "prediction".

Pp 11, Equation 1, format this equation in a conventional mathematical format, it is difficult to read in the manner that is presented.

Author Response

Thank you for your helpful revision, we tried to acknowledge as much as we could. We replied to every commentary in blue.

The manuscript presents the design of a low cost system that is able to monitor radon concentration levels as well as other variables of interest using several soft and hard technologies. Even though the approach is interesting and relevant from a technical point of view there is a main drawback with the manuscript: it lacks of a validation method. I strongly suggest to include a methodology to validate the performance in terms of standard metrics in order to evaluate both the security and the monitoring capabilities of the system. Without a validation method, the study is just a integration of several technologies that authors claim that works properly with no scientific evidence. It is difficult to test a system that is not intended to improve upon another system. We included two graphs in the results sections, comparing the radon levels in two scenarios for the same location with a particularly high radon concentration (we use it as one of our testing sites): one without the alert system (the system just stores the radon measures) and the other with the alert system activated and a human operator in charge of turning an air flow control system on or off. With this comparison we try to show that the system can help to significantly lower the radon levels. We included text explaining this. By the other hand, the title of the study does not reflect the fact that the system designed is a monitoring system not a control one. Authors do not present any control system issue developed in the study at all (open loop, closed loop). We changed the title of the article to: “Developing a secure low-cost radon monitoring system”. In order to enhance the content of the manuscript there are several considerations that I suggest to review: Pp 1, line 32 use "etc" or "so on" or "among others", instead of ellipsis (dots). Although we think that using “etc” could be acceptable (see Garner's Modern American Usage) we completed the sentences properly when possible to avoid its usage (and be more precise). Pp 3, line 50, “transforming the process” is confuse, due to the statement is related to radon transformation not other process transformation. Corrected. It was meant to say: “… and transforming, in the process, into...”. Pp 7, line 194, why approximately, an explanation is required. We added an explanation: “The system intends to continuously sample radon levels close to real-time: a real-time system would just detect a particle collision; this measure needs to be integrated over time (usually 10 minutes to 1 hour), extrapolated for a longer time frame and compared to the limit used in the system (regulatory limits are usually established for annual continuous levels)”. Pp 7, line 202, "said device" or "such device"?. We changed it to: “associated with the device”. Pp 9, Fig 4 is messy, please re edite using a CAD software such as Fritzing (looks like it was used). We replaced Fig 4 with a cleaner diagram (yes, it was done with Fritzing :-)). Pp 11, line 324, Fig 5. Corrected. Pp 12, line 333, it is more apropiate "estimation" instead of "prediction". Changed to estimation. Pp 11, Equation 1, format this equation in a conventional mathematical format, it is difficult to read in the manner that is presented. Corrected (we were using a more computer science oriented notation).

Reviewer 3 Report

The paper reports the development of a secure low-cost control system to detect Radon gas which is a severe health threat. There is no or very limited novelty of the work as it attempts to develop an application using existing tools and services, and does not propose a new concept or algorithm. The importance of the work, however, lies in its usage where it can make a difference. An application of this type can be of great value at places where Radon gas level could cause a health hazard.

The paper has all the potential to be a good paper, but in its current state, it is not of the standard to be accepted. I, therefore, propose a major correction before it gets accepted. My review will provide detailed comments as to how to improve the paper to reach that standard.

First and foremost, the paper requires one or more solid research questions. I can see those research questions floating around here and there in the text. After having a couple of reading of the paper, I realised that it attempts to solve two things:

i) It informs householders of the presence of radon gas by detecting its presence before reaching a dangerous level and ii) in doing so, secure the end-to-end communication from potential hackers to avoid foul plays.

I suggest the authors create a section called 'research problem' and frame the problem along with the research questions before attempting to solve it.

Second, the paper does not justify the work that the authors had done in the design and implementation parts of this project. Because of the work not presenting a novel idea or algorithm, it is of utmost necessity that the implementation work takes a research approach and explains the related work. It is important to show what had previously done and why the authors think that work is not sufficient or not a perfect fit for the use-case they have been dealing with. This will create the basis for the implementation described in Section 3. The section itself needs rewriting. The process presented at lines 304 and 315 can be well-presented in the form of pseudocodes.

Third, the paper took security lightly, whereas this could have been a strength of the work. Proper analysis of the potential security breaches using a threat model can significantly improve the quality; hence I suggest adding so. The authors seem to be jumping on to the solution before analysing and describing the problem to the readers, which is essential. Overall, I recommend making security a separate section and expand it with detailed explanations, attacks, adversaries and a threat model.

Forth, the result section of the paper is fragile. I anticipated a real comparison between the human recorded and the system detected radon level showing the importance of the latter. A trial of a couple of weeks would have been sufficient, but having it will significantly uplift the standard of the paper. The charts presented in this section are not readable. Regrettably, the authors did not consider creating the charts using a tool such as excel or such instead took a photograph of the web interface and included it (line 488). I strongly recommend presenting clear and readable charts in the next version of this submission.

Fifth, the paper includes images of low resolution that are difficult to read and comprehend. For example, the map presented at line 122 is of inferior quality. Authors are suggested to include a high-resolution version or remove the map altogether. In the case of latter, linking the map pointing its availability on the web will be sufficient.

Sixth, the paper requires editing and rewriting. For example, the first line of the introduction looks incomplete (look at line 32). Lines 47, 54, 60 and 74 contain descriptions and claims that require referencing.

Author Response

Thank you for your helpful revision, we tried to acknowledge as much as we could. We replied to every commentary in blue.

The paper reports the development of a secure low-cost control system to detect Radon gas which is a severe health threat. There is no or very limited novelty of the work as it attempts to develop an application using existing tools and services, and does not propose a new concept or algorithm. The importance of the work, however, lies in its usage where it can make a difference. An application of this type can be of great value at places where Radon gas level could cause a health hazard. The paper has all the potential to be a good paper, but in its current state, it is not of the standard to be accepted. I, therefore, propose a major correction before it gets accepted. My review will provide detailed comments as to how to improve the paper to reach that standard. First and foremost, the paper requires one or more solid research questions. I can see those research questions floating around here and there in the text. After having a couple of reading of the paper, I realised that it attempts to solve two things: i) It informs householders of the presence of radon gas by detecting its presence before reaching a dangerous level and ii) in doing so, secure the end-to-end communication from potential hackers to avoid foul plays. I suggest the authors create a section called 'research problem' and frame the problem along with the research questions before attempting to solve it. We moved all the text related to this to a “research problem” section and completed it, removing the “State of Art” section. Second, the paper does not justify the work that the authors had done in the design and implementation parts of this project. Because of the work not presenting a novel idea or algorithm, it is of utmost necessity that the implementation work takes a research approach and explains the related work. It is important to show what had previously done and why the authors think that work is not sufficient or not a perfect fit for the use-case they have been dealing with. This will create the basis for the implementation described in Section 3. We included some references to acknowledge this in the “research problem” section. The section itself needs rewriting. The process presented at lines 304 and 315 can be well-presented in the form of pseudocodes. We used pseudocode instead of text for those lines and rearranged the text of that part of the section. Third, the paper took security lightly, whereas this could have been a strength of the work. Proper analysis of the potential security breaches using a threat model can significantly improve the quality; hence I suggest adding so. The authors seem to be jumping on to the solution before analysing and describing the problem to the readers, which is essential. Overall, I recommend making security a separate section and expand it with detailed explanations, attacks, adversaries and a threat model. Security was already in a separate section, but maybe the level 2 title format doesn’t make it clear. We created a level 1 section for security and included a threat model graph and added some text to the section. Forth, the result section of the paper is fragile. I anticipated a real comparison between the human recorded and the system detected radon level showing the importance of the latter. A trial of a couple of weeks would have been sufficient, but having it will significantly uplift the standard of the paper. The charts presented in this section are not readable. Regrettably, the authors did not consider creating the charts using a tool such as excel or such instead took a photograph of the web interface and included it (line 488). I strongly recommend presenting clear and readable charts in the next version of this submission. We included two graphs in the results sections, comparing the radon levels in two scenarios for the same location with a particularly high radon concentration (we use it as one of our testing sites): one without the alert system (the system just stores the radon measures) and the other with the alert system activated and a human operator in charge of turning an air flow control system on or off. With this comparison we try to show that the system can help to significantly lower the radon levels. We included text explaining this. Fifth, the paper includes images of low resolution that are difficult to read and comprehend. For example, the map presented at line 122 is of inferior quality. Authors are suggested to include a high-resolution version or remove the map altogether. In the case of latter, linking the map pointing its availability on the web will be sufficient. We improved the images (legends, for instance) an included a high resolution version of them. Sixth, the paper requires editing and rewriting. For example, the first line of the introduction looks incomplete (look at line 32). Lines 47, 54, 60 and 74 contain descriptions and claims that require referencing. We corrected the text and added the references.

Round 2

Reviewer 2 Report

I consider that authors responded in a proper manner all the inquiries asked. They took into account all the comments with enough arguments.

Reviewer 3 Report

I appreciate the effort authors took to improve the paper. They addressed most of my comments that significantly enhanced the quality of the work. I am happy with the responses they made and believe the article is ready to go for publication subject to proofreading and addressing minor styling issues.